# Mortality of Patients Infected by COVID-19 with and without Deep-Vein Thrombosis

**DOI:** 10.3390/medicines8120075

**Published:** 2021-11-29

**Authors:** Jose Maria Pereira de Godoy, Gleison Juliano da Silva Russeff, Carolina Hungaro Costa, Debora Yuri Sato, Desirée Franccini Del Frari Silva, Maria de Fatima Guerreiro Godoy, Henrique Jose Pereira de Godoy, Paulo César Espada

**Affiliations:** 1Cardiovascular Surgery Department, Sao Jose do Rio Preto School of Medicine (FAMERP), Undergraduate Medicine Course and *Stricto Sensu* Postgraduate Course (FAMERP) and CNPq (National Council for Research and Development), São Jose do Rio Preto 15025120, Brazil; 2Vascular Surgery Echography Service, Hospital Affiliated with Sao Jose do Rio Preto School of Medicine (FAMERP), São Jose do Rio Preto 15025120, Brazil; gjrusseff@hotmail.com (G.J.d.S.R.); carolinacunha1@gmail.com (C.H.C.); deboraysato@hotmail.com (D.Y.S.); dfrancinni@yahoo.com.br (D.F.D.F.S.); 3*Stricto Sensu* Postgraduate Program, Sao Jose do Rio Preto School of Medicine (FAMERP), Research Group at Godoy Clinic, Sao Jose do Rio Preto 15025120, Brazil; mfggodoy@gmail.com; 4Department of General Surgery, Sao Jose do Rio Preto School of Medicine (FAMERP), São Jose do Rio Preto 15025120, Brazil; henriquegodoy95@gmail.com; 5Trauma Surgery, Sao Jose do Rio Preto School of Medicine (FAMERP), São Jose do Rio Preto 15025120, Brazil; pespada@terra.com.br

**Keywords:** COVID-19, mortality, deep-vein thrombosis, Doppler, ultrasound

## Abstract

Background: Current evidence points to a state of hypercoagulability (consequence of hyperinflammation) as an important pathogenic mechanism that contributes to the increase in mortality in cases of COVID-19. The aim of the present study was to investigate the influence of deep-vein thrombosis on mortality patient’s infected with SARS-CoV-2. Method: A clinical trial was conducted involving 200 consecutive patients with COVID-19—100 patients who were positive for deep-vein thrombosis (venous Doppler ultrasound) and 100 who were negative for deep-vein thrombosis at a public hospital. Results: The mortality rate was 67% in the group positive for DVT and 31% in the group negative for DVT. Conclusion: Deep-vein thrombosis is associated with an increase in mortality in patients with COVID-19 and failures can occur with conventional prophylaxis for deep-vein thrombosis.

## 1. Introduction

Current evidence points to a state of hypercoagulability (a consequence of hyperinflammation) as an important pathogenic mechanism that contributes to the increase in mortality in cases of COVID-19. This theory is supported by reports of high inflammatory markers and clotting as well as a correlation between high interleukin-6 (IL-6) levels and fibrinogen. The anti-inflammatories with anticoagulants could decrease thrombotic events and related fatal consequences [1,2,3,4].

An autopsy study found that 81% of patients with COVID-19 as the main cause of death exhibited thrombotic phenomena in the lungs. The extrapulmonary thrombotic events were ischemic stroke (N, % = 4; 3.95), acute myocardial infarction (N, % = 3; 2.94), and critical lower limb ischemia (N, % = 1; 0.98) [5]. Another study showed that autopsy revealed deep-vein thrombosis in (58%) in whom venous thromboembolism was not suspected before death [6].

The underlying mechanisms proposed for these severe manifestations involve immunological dysregulation, including an antiphospholipid syndrome-like state, activation of the complement system, viral dissemination with direct systemic endothelial infection, viral RNAemia with immunothrombosis, activation of the clotting pathway mediated by hypoxemia, and immobility [7,8,9,10]. Antiphospholipid syndrome is associated with arterial and venous thrombosis, but other causes of congenital thrombophilia should be considered, such as the deficiencies in C protein, S protein, antithrombin III, etc. [11,12]. Arterial and venous thrombotic conditions are among the most serious complications of COVID-19, and the diagnosis of these conditions using Doppler ultrasound is the main form of evaluation of the extremities [13,14]. Thromboprophylaxis is mainly performed with low-molecular-weight heparin (LMWH) and, in some specific patients, with unfractionated heparin (UFH) [15].

The aim of the present study was to investigate the influence of deep-vein thrombosis on mortality patient’s infected with SARS-CoV-2.

## 2. Methods

### 2.1. Patients and Setting

A total of 200 consecutive patients with COVID-19 (100 with DVT and 100 without DVT) were evaluated at the public hospital affiliated with the São Jose do Rio Preto School of Medicine (SP, Brazil) between March 2020 and May 2021. Deep-vein thrombosis (DVT) was investigated using venous Doppler ultrasound.

### 2.2. Design

A clinical trial was conducted involving 100 patients with COVID-19 positive for DVT (venous Doppler ultrasound) and 100 negative for DVT. Patients negative for COVID-19 and those with inconclusive exams were excluded. The difference in mortality was analyzed using Fisher’s exact test.

### 2.3. Inclusion and Exclusion Criteria

All patients with COVID-19 submitted to venous Doppler ultrasound at the public hospital affiliated with the São Jose do Rio Preto School of Medicine with a suspicion of DVT were included in the study. Patients negative for COVID-19 and those with inconclusive exams were excluded.

### 2.4. Ethical Considerations

This study received approval from the institutional review board of the São Jose do Rio Preto School of Medicine, SP, Brazil #4.720.521. All participants signed a statement of informed consent.

### 2.5. Statistical Treatment

Descriptive statistics were performed on the data and comparisons were made using Fisher’s exact test, considering an alpha error of 5%.

### 2.6. Selection of Patients

Consecutive patients with COVID-19 submitted to venous Doppler ultrasound until reaching 100 positive for DVT and 100 negative for DVT.

### 2.7. Development

The São Jose do Rio Preto Hospital had 5559 patients with COVID-19 during the study: 3706 (71.8%) in the medical wards and 1453 (28.1%) in the intensive care units. More than 200 patients in the ICUs had a suspicion of DVT and were sent for bilateral venous Doppler ultrasound of the lower limbs. These patients were subsequently divided into two groups: 100 positive for DVT and 100 negative for DVT. The mean D-dimer results, age group, and mortality were evaluated in each group. The data were entered into an Excel table and analyzed descriptively. Comparisons between groups were performed using Fisher’s exact test, considering a 5% alpha error.

## 3. Results

Mean age was 55.58 ± 12.58 years in the group positive for DVT and 58.61 ± 14.2 years in the group negative for DVT. This difference was not statistically significant (*p* = 0.1, paired *t*-test). The mortality rate was 67% in the group positive for DVT and 31% in the group negative for DVT. This difference was statistically significant (*p* = 0.0001, Fisher’s exact test) (Table 1). The mean D-dimer level was 11.9 ± 7.15 um in the group with DVT and 4.97 ± 5.3 um in the group without DVT. The median difference was 10.25 (CI: 8.51 to 13.46) and was statistically significant (*p* < 0.0001, Mann–Whitney U test) (Figure 1).

## 4. Discussion

The present study points to deep-vein thrombosis (DVT) as one of the main complications of SARS-CoV-2 related to mortality. The mortality rate among the patients with DVT was more than double that found in the group without DVT. The patients were from a single teaching school, where the conduct in the intensive care units is uniform and all patients received injectable antithrombotic prophylaxis. This prophylactic procedure was based on changes in D-dimer levels and clinical aspects. The literature offers several studies reporting pulmonary thrombosis in approximately 80% of COVID-19-related deaths [11], but no previous study has evaluated the influence of DVT.

One of the characteristics of COVID-19 is the involvement of multiple sites of micro- and macro-thrombosis in the lungs. Venous thrombotic events of the lower limbs are also seen [16,17,18]. However, these details are not part of the present study. The characteristics of thrombotic events differ from those found in patients without COVID-19. Therefore, a novel presentation characteristic of thrombotic events is seen in this pandemic. A study has shown that the mortality rate was significantly higher (16.0%) in patients with thrombosis than in those without thrombosis (7.9%) [19]. The vast majority of individuals who died directly from SARS-CoV-2 infection were of advanced age and had multiple comorbidities [20].

One hypothesis for the increase in mortality is the greater occurrence of extrapulmonary thrombosis. Micro- and macro-thrombotic events can affect the entire organism, thereby increasing the sequelae of this physiopathological process [17]. The D-dimer level practically doubles in comparison to patients without DVT, suggesting a greater volume of lysis due to the thrombotic process.

Some patients without DVT had high D-dimer levels, suggesting thrombotic involvement in other parts of the body, such as the upper limbs and abdomen. Arterial thrombotic events have been seen at our service, but with less frequency. The early diagnosis of a thrombotic event is fundamental so that these patients can receive adequate treatment for DVT.

While prophylaxis for DVT is fundamental, failures in prevention have occurred even when using Clexane 40 um every 12 h, suggesting considerable viral aggression to the vascular system.

## 5. Conclusions

Deep-vein thrombosis is associated with an increase in mortality in patients with COVID-19, and failures can occur with conventional prophylaxis for deep-vein thrombosis.

## Figures and Tables

**Figure 1 medicines-08-00075-f001:**
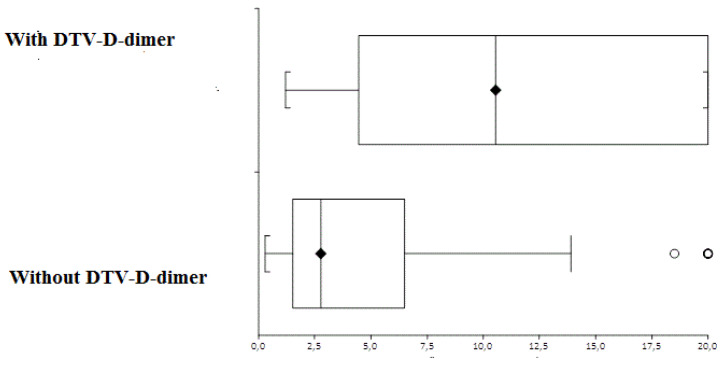
Median D-dimer levels and interquartile ranges in patients with and without deep-vein thrombosis.

**Table 1 medicines-08-00075-t001:** Descriptive statistics of patients with and without deep-vein thrombosis.

Variables	Without DVT	With DVT
Valid data	100	100
Mean	3.59	7.47
Standard deviation	3.58	28.94
Median	2.33	3.3

## Data Availability

The data used to support the findings of this study are included within the article.

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
