# Peer review of "Mortality of Patients Infected by COVID-19 with and without Deep-Vein Thrombosis"

_medicines, 2021, doi:10.3390/medicines8120075_

Round 1
Reviewer 1 Report
This paper discusses the influence of deep vein thrombosis on the mortality rate of patients infected with COVID-19. I believe that the authors must develop more the sections of this paper. Please see my comments in the document attached.

Author Response
REVIEWER 1
Current evidence points to a state of hypercoagulability 37 (consequence of hyperinflammation) as an important pathogenic 38 mechanism that contributes to the increase in mortality in cases of 39 COVID-19.
This theory is supported by reports of high inflammatory 39 markers and clotting as well as a correlation between high interleukin-40 6 (IL-6) levels and fibrinogen.1
1) Please try to rewrite the introduction!! lacks of inforamtion and story of this article!!
Reply: ok, made.
2) Please develop this idea within a paragraph!!
Reply: Ok.
3) Add more reference
Reply: Ok, made.
Patients and Setting
Two hundred, consecutive patients with COVID-19 were evaluated at the public hospital affiliated with the São Jose do Rio Preto School of Medicine (SP, Brazil) between March 2020 and May 2021. Deep vein thrombosis (DVT) was investigated using venous Doppler ultrasound.
not the same number at the abstract!? please check it!!
Reply: Two hundred, 100 with DVT, 100 without DVT.
Did you get any kind of authorization from the patients for this kind of study??
Reply: Yes the study was approved Ethical Committee and consent form signed.
Results
describe more this figure!!
Reply: The authors improving figure 1, 300DPI. No is necessary describing figure, in study show interpretative results.
Discussion
need of other studies for comparison!!
Reply: ok, made.
Reviewer 2 Report
The paper is very short, especially the discussion section. It does not include citations from line 126, where the citations are required.
In order to acquire more scientific value English editing services are required.
line 60: “in patients infected by COVID-19” -> COVID-19 is a disease, SARS-CoV-2 is the etiology, I would suggest to change the sentence into: patient’s infected with SARS-CoV-2 or in COVID-19 patients
In the results section I would suggest not to write what tests were used in the statistical analysis considering the data type, but to include this information in the methods section
Author Response
REVIEWER 2
The paper is very short, especially the discussion section. It does not include citations from line 126, where the citations are required.
Reply: Include new citation in discussion.
In order to acquire more scientific value English editing services are required.
Reply: The version was editing professional native English. Attached declaration editing.
line 60: “in patients infected by COVID-19” -> COVID-19 is a disease, SARS-CoV-2 is the etiology, I would suggest to change the sentence into: patient’s infected with SARS-CoV-2 or in COVID-19 patients
Reply: Ok, change in text.
In the results section I would suggest not to write what tests were used in the statistical analysis considering the data type, but to include this information in the methods section.
Reply: Not agree, in results is important refer type statistic test, the authors write test in statistical analysis too.
Reviewer 3 Report
The authors describe the impact of a DVT diagnosed by venous Doppler ultrasound on mortality and compare it to negative results.
The conclusions driven from the data are sound and significant. Moreover, necessary preconditions like homogeneous treatment in both groups are described before.
Doppler sound results were conclusive with D-Dimer results. In Autopsies, the elevated frequency of thromboembolic conditions in the patients who died was objectively verified.
This study has a clear concept and precise results!
Author Response
REVIEWER 3
The authors describe the impact of a DVT diagnosed by venous Doppler ultrasound on mortality and compare it to negative results.
The conclusions driven from the data are sound and significant. Moreover, necessary preconditions like homogeneous treatment in both groups are described before.
Doppler sound results were conclusive with D-Dimer results. In Autopsies, the elevated frequency of thromboembolic conditions in the patients who died was objectively verified.
This study has a clear concept and precise results!
Reply: Thanks so much for consideration.
Reviewer 4 Report
The manuscript is generally well-written and the results are properly presented. Figure 1 definition is not really good, so I suggest improving its quality.
Author Response
REVIEWER 4
The manuscript is generally well-written and the results are properly presented. Figure 1 definition is not really good, so I suggest improving its quality.
Reply: The authors improving figure 1, 300DPI.
Round 2
Reviewer 1 Report
I would like to thank the authors for their modifications which improve the quality of the document. But, I am a bit worried about the Introduction which needed more improvement and more information and references. I am suggesting that the authors improve the Introduction before processing this paper for publication.
Author Response
I would like to thank the authors for their modifications which improve the quality of the document. But, I am a bit worried about the Introduction which needed more improvement and more information and references. I am suggesting that the authors improve the Introduction before processing this paper for publication.
Reply: Ok, made.
Reviewer 2 Report
Line 51-53, please present the results this way: n/N, %, instead of n,%
line 82-83, please change the inclusion criteria into a sentence, after "DVT" write that these patients were included in the study.
line 86-86, please change that part into a sentence, ending with: these patients were excluded
Author Response
Line 51-53, please present the results this way: n/N, %, instead of n,%
Reply: Ok, made.
line 82-83, please change the inclusion criteria into a sentence, after "DVT" write that these patients were included in the study.
Reply: Ok, made.
line 86-86, please change that part into a sentence, ending with: these patients were excluded
Reply: Ok.